# Qualitative Analysis and Componential Differences of Chemical Constituents in Lysimachiae Herba from Different Habitats (Sichuan Basin) by UFLC-Triple TOF-MS/MS

**DOI:** 10.3390/molecules27144600

**Published:** 2022-07-20

**Authors:** Yongyi Zhou, Haijie Chen, Jia Xue, Jiahuan Yuan, Zhichen Cai, Nan Wu, Lisi Zou, Shengxin Yin, Wei Yang, Xunhong Liu, Jianming Chen, Fushuangshuang Liu

**Affiliations:** 1College of Pharmacy, Nanjing University of Chinese Medicine, Nanjing 210023, China; zyy18851091328@163.com (Y.Z.); chenhaijie039x@163.com (H.C.); 17860505214@163.com (J.X.); yuanjiahuan1027@163.com (J.Y.); caizhichen2008@126.com (Z.C.); wunan7272@163.com (N.W.); yinshengxin723@163.com (S.Y.); yangwei202103@163.com (W.Y.); cjm7895@163.com (J.C.); 18801588692@163.com (F.L.); 2Jiangsu Province Engineering Research Center of Classical Prescription, Nanjing 210023, China

**Keywords:** Lysimachiae Herba, habitats, chemical constituents, UFLC-triple TOF-MS/MS

## Abstract

Lysimachiae Herba (LH), called Jinqiancao in Chinese, is an authentic medical herb in Sichuan Province often used in the prescription of traditional Chinese medicine (TCM). However, in recent years, there has been a lack of comprehensive research on its chemical components. In addition, the landform of Sichuan Province varies greatly from east to west and the terrain is complex and diverse, which has an important influence on the chemical constituents in LH. In this study, ultrafast liquid chromatography coupled with triple-quadrupole time-of-flight tandem mass spectrometry (UFLC-triple TOF-MS/MS) was used to analyze the samples of LH from eight different habitats in Sichuan Basin. The constituents were identified according to the precise molecular weight, the fragment ions of each chromatographic peak and the retention time of the compound obtained by high-resolution mass spectrometry, combined with software database searches, standard comparisons and the related literature. Differential chemical constituents were screened using partial least squares discriminant analysis (PLS-DA) and *t*-tests. The results showed that a total of 46 constituents were identified and inferred, including flavonoids, phenolic acids, amino acids, tannins, fatty acids and coumarins; the fragmentation pathways of the main constituents were preliminarily deduced. According to the variable importance in projection (VIP) and *p*-values, four common differential constituents were screened out, 2-*O*-galloylgalactaric acid, quercetin 3-*O*-xylosyl-rutinoside, nicotiflorin and kaempferol 3-rutinosyl 7-*O*-alpha-l-rhamnoside. This study provides basic information for the establishment of a comprehensive quality evaluation system for LH.

## 1. Introduction

Lysimachiae Herba (LH), which is the dried whole herb of *Lysimachia christinae* Hance., has the effect of promoting diuresis and removing jaundice, along with anti-inflammatory and analgesic properties. It is commonly used for symptoms such as jaundice, hypochondriac pain and urolithiasis [1] and it is well known as a key medicine for the treatment of lithiasis. Hitherto, many studies have shown that LH has a variety of pharmacological effects such as the promotion of bile secretion, as well as anti-inflammation, analgesic, bacteriostatic and anti-gout properties [2]. Chemical constituents are the basis for the pharmacological action of traditional Chinese medicine (TCM). Phytochemical analyses have revealed that LH contains multiple chemical constituents, such as flavonoids [3,4,5], phenolic acids [4] and fatty acids [5]. In fact, the quality standard of LH recorded in the Chinese Pharmacopoeia (2020 version) mainly involves the quantification of quercetin and kaempferol [1] and there is still a lack of research on the other constituents of LH. Therefore, it is necessary and important to study its chemical constituents systematically.

It is commonly believed that genuine Chinese herbs refer to medicinal materials produced in a specific area of natural conditions. The production is relatively concentrated with certain cultivation techniques and harvesting and processing methods, with excellent quality and effect, as recognized by the clinical practice of TCM. LH is distributed in provinces south of the Yangtze River and Taiwan, but the most well-known habitat is Sichuan Province. Sichuan Province is located inland in southwest China and the upper reaches of the Yangtze River and its terrain is characterized by being high in the west and low in the east. The climate of Sichuan Province is affected by the topography, which can be roughly divided into two climatic zones: Sichuan Plateau and Sichuan Basin [6]. In this study, samples from eight habitats in Sichuan Basin were analyzed to systematically study the chemical constituents of LH and distinguish their differences in different habitats. The research results can preliminarily clarify the material basis of changes in different ecological environments and provide information for rational clinical application.

In recent years, plant metabolomics and liquid chromatography/mass spectrometry (LC-MS) have been widely used in the analysis of complex systems of TCM. Plant metabolomics is a technology for the comprehensive analysis of metabolites in plants, which is especially suitable for multicomponent system analysis of TCM [7]. Combined with the advantages of the high separation of the liquid phase and high sensitivity of mass spectrometry, LC-MS can be used to separate and analyze complex samples and identify their structures. It has been widely used in quantitative and qualitative analysis of complex systems of TCM. Principal component analysis (PCA) is a statistical method that uses the concept of dimension reduction to recombine variables into new composite variables, from which a small number of variables are selected to reflect most of the original information [8]. Partial least squares discriminant analysis (PLS-DA) commonly uses the variable importance in projection (VIP) value to describe the degree of contribution of a variable, which can be considered significant when VIP > 1. Among them, *Q*^2^ > 0.5 indicates that the predicted value is high, while as *R*^2^*X* and *R*^2^*Y* approach 1, the model becomes more stable. In this study, a total of 46 chemical constituents in LH were screened out, most of which were flavonoids and phenolic acids. According to the VIP obtained by PLS-DA (VIP > 1) and *t*-tests (*p* < 0.05), characteristic peaks of the differentiated chemical constituents were screened out and four commonly differential chemical constituents were finally identified. According to the research ideas and methods of plant metabolomics, this experiment used UFLC-triple TOF-MS/MS to analyze the chemical constituents in LH and combined multivariate statistical analysis to explore the variations in chemical constituents in LH from different areas in Sichuan Basin. Therefore, this study lays a certain foundation for the basic research and quality control of LH medicinal substances.

## 2. Results

### 2.1. Optimization of Extraction Conditions

This experiment investigated the single factors of the extraction solvent (60%, 70%, 80%, 90% and 100% methanol; 60%, 70%, 80%, 90% and 100% ethanol), extraction time (15, 30, 45, 60, 75 and 90 min) and solid–liquid ratio (1:10, 1:20, 1:30, 1:40 and 1:50, *w*/*v*). The results showed that more substances could be detected using 80% methanol solvent, a 1:10 solid–liquid ratio and 60 min ultrasonic extraction. Since no smaller proportions of the solid–liquid ratio of 1:10 were investigated, the five solid–liquid ratios of 1:6, 1:8, 1:10, 1:12 and 1:14 were reconfigured and ultrasonically extracted using 80% methanol for 60 min. In the end, the 1:10 solid–liquid ratio was found to be more efficient. Therefore, 80% methanol at room temperature, a 1:10 solid–liquid ratio and 60 min ultrasonic extraction were selected as the optimal extraction conditions.

### 2.2. Optimization of UFLC-Triple TOF-MS/MS Conditions

The effects of methanol–0.1% (*v*/*v*) formic acid in water, methanol–water, acetonitrile–0.1% (*v*/*v*) formic acid in water and acetonitrile–water as mobile phases under gradient elution on the separation of the peaks in the samples were investigated and we found the methanol–0.1% formic acid in water solution as the mobile phase to achieve a good separation effect. In addition, compared with the electrospray positive ion mode, the negative ion mode was found to have a higher LC/MS response; hence, the samples were measured in negative ion mode in the experiment.

### 2.3. Identification of the Constituents in LH

According to the corresponding chromatographic and mass spectrometric conditions, the chemical constituents of LH samples from eight habitats were identified. The results showed that the constituents identified in the samples from Sichuan Bazhong were more comprehensive. Figure 1 shows the base peak chromatogram (BPC) of the LH sample from Sichuan Bazhong in negative ion mode. A total of 46 constituents were identified, including 25 flavonoids, 11 phenolic acids and 10 other constituents. The detailed information of the identified compounds is shown in Table 1, with their corresponding structures in Figure 2.

#### 2.3.1. Identification of Flavonoids and Their Glycosides

Flavonoids are the main constituents identified in LH and the main active ingredients of LH [2]. Natural flavonoids are flavonoid glycosides with 15-carbon [A(C_6_)-C(C_3_)-B(C_6_)] as the parent nucleus and their derivatives. Their basic parent nuclei are diphenyl chromogenic ketones, mostly in the form of glycosides. The cleavage of flavonoid glycosides is characterized by the breakage of glycosidic bonds, sugar–glycan bonds and the cross-loop excision of sugar rings. A total of 25 flavonoids were identified in this study, including flavonols, flavonoids, dihydroflavonoids, flavonols, chalcones and flavonoid glycosides. Several retro-Diels–Alder (RDA) cleavage modes of flavonoids and flavonols are shown in Figure 3.

Flavonol compounds are prone to lose neutral fragments CO (28 Da), CO_2_ (44 Da), H_2_O (18 Da) and C_2_H_2_O (42 Da), as well as undergo RDA reactions in the C ring (process where a six-membered cyclic compound with a double bond is bombarded by an ion source and decomposed into diene and pro-diene fragment products). Taking compound **44** as an example, its excimer ion peak *m*/*z* 301.0348 [M − H]^−^ removed a CO and produced a fragment ion at *m*/*z* 273.0393. Then, the C ring underwent 1,2 and 1,3 bond cleavage to produce the fragment ion at *m*/*z* 179.0341 [^1,^^2^A]^−^ and *m*/*z* 151.0034 [^1,3^A]^−^, followed by the removal of a CO_2_ to produce the fragment ion at *m*/*z* 107.0172; the cleavage pathway is shown in Figure 4A. Considering the literature and compared with the standard, compound **44** was identified as quercetin. Due to the methoxy at the 4′ position, compound **43** only produced simple fragmentation at *m*/*z* 284.0307 [M − H − CH_3_]^−^ in secondary mass spectrometry. Considering the literature, it could be tentatively identified as kaempferide.

Flavonol glycosides preferentially lose the glycosyl fraction in mass spectrometric collisions and the ions can further generate RDA cleavage fragments. Compounds **24**, **26**, **27**, **31**, **33**, **34**, **35**, **36**, **37**, **38**, **39** and **40** were identified as flavonol glycosides, among which **24**, **27**, **34**, **35**, **36** and **39** were quercetin-based flavonol glycosides, while the others were kaempferol-based flavonol glycosides. As shown in Figure 4B, the excimer ion peak *m*/*z* 609.1461 [M − H]^−^ of compound **35** lost one molecule of rutinose and produced the fragment ion *m*/*z* 301.0353, which then underwent further RDA cleavage of the C ring to produce the fragment ion *m*/*z* 151.0035. Accordingly, *m*/*z* 151 can be used as the characteristic ion for quercetin-based flavonol. Considering the literature and compared with the standard, compound **35** was identified as rutin. Flavonol glycosides with kaempferol as the parent nucleus generally produce the [^1,3^A]^−^ fragment ion at *m*/*z* 151.0044 and another *trans*-ring cleavage [^0,^^2^A]^−^ ion at *m*/*z* 162.8200, with a large number of ions after the loss of neutral fragments.

Flavonoids undergo frequent loss of neutral fragments such as CO, CO_2_, H_2_O, CH_3_ (15 Da) and OCH_3_ (31 Da) in negative ion mode and, in contrast to flavonols, the C ring of flavonoids does not open readily. Flavonoids mainly undergo RDA cleavage and produce two fragment ions, [^1,3^A]^−^ (151 Da) and [^1,3^ B]^−^ (133 Da). Compounds **22**, **23**, **29**, **30**, **32** and **42** were classified as flavonoids. For flavonoid compounds containing the hydroxyl on the B ring (such as luteolin and apigenin), compounds containing only 4′ hydroxyl are more likely to produce [^1, 3^A]^−^ fragment ions than compounds containing 3′,4′ dihydroxy. Taking compound **22** as an example, its excimer ion peak at *m*/*z* 269.0564 [M − H]^−^ produced fragment ions *m*/*z* 117.0345 [M − H − C_7_H_4_O_4_]^−^ and *m*/*z* 151.0072 [^1,3^A]^−^ after RDA cleavage; compared with the standard, it could be tentatively identified as apigenin.

Compounds **23**, **29**, **30** and **32** were classified as flavonoid *C*-glycosides (where the end-group carbon atom of the sugar group is directly linked to the glycosidic carbon atom). The cleavage pattern of flavonoid carbohydrates is regular. If the fragmentation peaks of the deglycosylated group do not appear first, the following regular fragment ions are present in the MS-MS spectra of flavonoid carboxides: [M − H − C_2_H_4_O_2_]^−^, [M − H − C_3_H_6_O_3_]^−^ and [M − H − C_4_H_8_O_4_]^−^; hence, it can be basically concluded that they are six-carbon flavonoid *C*-glycosides. Taking compound **29** (*m*/*z* 431.0982 [M − H]^−^) as an example, the ion peak removed mass numbers of **28** (formyl) and **30** (methoxy), revealing the MS fragmentation pattern of the carbanion. Combined with the fragmentation pattern and structural features of known compounds, this compound could be identified as a flavonoid *C*-glycoside with the 2-positions of glucose substituted by formyl and methoxy. After comparison with the standard, compound **29** could be identified as vitexin.

Compound **11** was epigallocatechin, a catechin compound, with a large number of breaks between the 1- and 2-, 3- and 4-bonds in the C ring, resulting in ion fragments such as 167.0339 [^1,2^A]^−^, 137.0266 [^1,3^A]^−^ and 125.0275 [^1,4^A]^−^. Compound **46** was identified as naringenin, a dihydroflavonoid. Compound **16** was identified as eriocitrin, a di-glycoside with dihydroflavonoid as the parent nucleus, with ring-opening breakage of the C ring C_2_–O_1_ and C_4_–C_10_ bonds after removal of the glycoside. Compound **17**, a monoglycoside with chalcone as the parent nucleus, was presumed to be marein after literature review.

#### 2.3.2. Identification of Phenolic Acids

Phenolic acids refer to aromatic carboxylic acid compounds with multiple phenolic hydroxyl substitutions on the benzene ring. They generally have a high response in negative ion mode, where the loss of neutral molecules CO_2_, H_2_O and CO mainly occurs to produce the ion fragments [M − H − CO_2_]^−^, [M − H − H_2_O]^−^ and [M − H − CO]^−^. When the structure of the compound contains caffeic acid, there is a tendency to lose caffeoyl and produce [M − H − caffeoyl]^−^ and caffeic acid fragment ions; when it contains gallic acid, there is a tendency to lose gallic acid and produce [M − H − gallic acid]^−^ and gallic acid fragment ions.

A total of 11 phenolic acids were identified in this experiment, namely, compounds **7**, **10**, **12**, **13**, **14**, **15**, **19**, **20**, **21**, **25** and **28**. The excimer ion peak for compound **7** was *m*/*z* 169.0143 [M − H]^−^, with the removal of one molecule of CO_2_ to produce the fragment ion at *m*/*z* 125.0248 and the sequential removal of two CO to produce the fragment ion at *m*/*z* 97.0322 and *m*/*z* 69.0390. Its cleavage pathway is shown in Figure 4C. Compound **7** was confirmed as gallic acid through comparison with the standard. The excimer ion peak of compound **10** (identified as protocatechuic acid) was stripped of a CO_2_ to produce the characteristic fragment ion *m*/*z* 109.0298 [M − H − CO_2_]^−^ in secondary mass spectrometry; the excimer ion peak of compound **19** (identified as caffeic acid) was stripped of a molecule of CO_2_, CO and H_2_O to produce the characteristic fragment ions at *m*/*z* 135.0449 [M − H − CO_2_]^−^, *m*/*z* 109.0440 [M − H − CO_2_ − CO]^−^ and 89.0413 [M − H − CO_2_ − CO − H_2_O]^−^, respectively, in secondary mass spectrometry.

Compound **15**, with an excimer ion peak of *m*/*z* 353.0872 [M − H]^−^, had the characteristic quinic acid fragment ion at *m*/*z* 191.0596, indicating that the compound often lost quinic acid (*m*/*z* 192) and produce caffeoyl (*m*/*z* 161) in the mass spectrometric cleavage. The cleavage pathway is shown in Figure 4D. On the basis of its mode of breakage and comparison with the standard, compound **15** was judged to be chlorogenic acid.

#### 2.3.3. Identification of Amino Acids

Amino acids are a class of organic compounds containing amino and carboxyl groups. In negative ion mode, the secondary mass spectra show fragmentation peaks of [M − H − NH_3_]^−^ and [M − H − CO_2_]^−^, which may be generated by the loss of NH_3_ and CO_2_ from amino acids. The three amino acids were identified mainly by comparison with the standards, according to the following process: firstly, we used Peakview to find the retention times of the corresponding amino acids in the mixed standard solution; secondly, we compared the *m*/*z* of substances with similar retention times (within 0.5 min) in the samples and those with *m*/*z* errors of more than 5 ppm were removed; then, the MS/MS patterns of the eligible amino acids consistent with the references and relevant web queries were determined. In the end, compounds **1**, **2** and **9** were presumed to be threonine, glutamic acid and phenylalanine, respectively. The comparison of the chromatograms of the three amino-acid standards with those of the extract is shown in Figure 5.

#### 2.3.4. Identification of Tannins

The basic structure of condensed tannins consists of flavan-3-ols such as (+)-catechin, (−)-epicatechin, or flavan-3,4-diols condensed by C−C at the 4,8- or 4,6-positions. The process of identification was as follows: first, the results of the analysis were imported into Peakview; then, compounds that had a mass error of less than 5 ppm, had the correct isotopic distribution and contained secondary fragments were identified as targets. Combining features of Peakview such as Formula Finder, matching the mass spectrometry data of each chromatographic peak in the database (SciFinder and HMDB) and considering the cleavage law of each peak, compounds **6** and **8** were eventually identified as prodelphinidin B1 and procyanidin B1. Taking compound **8** as an example, three cleavage pathways are postulated: first, the loss of a polymerization unit produced a fragment *m*/*z* 289.0749 [M − H − C_15_H_12_O_6_]^−^; second, an RDA reaction occurred and produced a neutral structure with a molecular of C_8_H_8_O_3_ (152 Da), while the fragment may have also further lost a molecule of H_2_O, producing an ion fragment *m*/*z* 407.0831 [M − H − C_8_H_8_O_3_ − H_2_O]^−^; third, a molecule of H_2_O and a fragment of 3,4-dihydroxybenzene were successively lost, producing an ion fragment *m*/*z* 451.0986 [M − H − H_2_O − C_6_H_5_O_2_]^−^. The MS^2^ spectra and speculated fragmentation pathways of procyanidin B1 are shown in Figure 6.

#### 2.3.5. Identification of Fatty Acids

Fatty acids are a class of carboxylic acid compounds that consist of hydrocarbon groups attached to carboxyl groups. Fatty acids have a better response in negative ion mode. Saturated fatty acids break the C_2_−C_3_ bond through γ-hydrogen migration and Mackenzie rearrangement to produce fragments with high intensity, while other fragments are generally (CH_2_)_n_−COOH with a 14n difference in relative molecular mass. Reviewing the literature, compounds **3**, **4**, **5** and **41** were tentatively presumed to be d-mannuronic acid, 2-deoxypentanoic acid, malic acid and 3-methylnonanoic acid. Taking compound **4** as an example, the secondary fragment ions *m*/*z* 73.0329, 89.0273 and 105.0218 corresponded to fragments [M − H − H_2_O − CO_2_ − CH_2_]^−^, [M − H − H_2_O − C_3_H_6_]^−^ and [M − H − CO_2_]^−^, respectively. The MS^2^ spectra and speculated fragmentation pathways of 3-methylnonanoic acid are shown in Figure 7.

#### 2.3.6. Identification of Coumarins

Coumarins have a lactone structure and can be structurally viewed as *cis*-*o*-hydroxycinnamates formed by intramolecular dehydration of the ring. Oxygen-containing functional groups can be substituted at various positions on the benzene ring, commonly including hydroxyl, methoxy and sugar groups. Compound **18** had fragments at *m*/*z* 177.0221 [M − H]^−^ in negative mode, with a major secondary fragment at *m*/*z* 149.0267 reduced by 28 from *m*/*z* 177.0221, presumably caused by the loss of CO, which is characteristic of the cleavage of coumarins. Therefore, it could be presumed that compound **18** was 6,7-dihydroxycoumarin and its MS^2^ spectrum and speculated fragmentation pathways are shown in Figure 8.

### 2.4. Analysis of the Differential Constituents of LH from Different Habitats

#### 2.4.1. PCA of the Samples

PCA was used to analyze the differences between the samples of LH from different habitats (S1, Bazhong; S2, Guangyuan; S3, Yibin; S4, Zigong; S5, Deyang; S6, Suining; S7, Leshan; S8, Nanchong) and the correlation between samples. The PCA model parameters of *R*^2^*X* = 0.865 and *Q*^2^ = 0.785 showed that the models were effective and reliable. As shown in Figure 9, the samples of LH from the eight habitats were clustered into one category and the distribution results were relatively ideal. Among them, the samples of S4 and S7 were highly aggregated and had little difference. The relative dispersion of S3 and other habitats indicated that the chemical constituents of S3 differed significantly from the samples of other habitats.

#### 2.4.2. PLS-DA of the Samples

In this experiment, the samples from the other seven habitats were compared with the samples from Bazhong and analyzed by PLS-DA. The results are shown in Figure 10a. As can be seen from the Figure 10, the samples from each habitat were obviously separated from S1 samples along the PC1 axis. The models were tested with 200 permutations and the results are listed in Table 2. The results showed that the models did not overfit, indicating that they were effective and reliable. According to the VIP score chart (Figure 10b) and *t*-tests corresponding to the model, differential chemical constituents (VIP > 1) of samples from different habitats in Sichuan Basin were screened out and the number of characteristic peaks is shown in Table 2.

#### 2.4.3. Identification of the Differential Chemical Constituents

A total of four common differential chemical constituents, 2-*O*-Galloylgalactaric acid, quercetin 3-*O*-xylosyl-rutinoside, kaempferol 3-*O*-rutinoside and kaempferol 3-rutinoside 7-rhamnoside, were identified from the eight samples of LH from different habitats in Sichuan Basin. The peak area of each common differential constituent was used as its relative content. The average value and standard deviation of the peak area of the same chemical constituent in different samples were calculated to obtain the relative content changes of common differential constituents between different samples (Figure 11). As shown in the figure, the content of all four differential constituents of LH from S1 was high.

## 3. Discussion

At present, there are few studies on the components of LH. In the current national standard, the quality control indicator of LH is the content of two glycosidic constituents, quercetin and kaempferol, which have small polarity. This study used UFLC-triple TOF-MS/MS to comprehensively analyze the chemical constituents of LH. Finally, 46 chemical constituents in LH were identified (Table 1), including flavonoids, phenolic acids and amino acids. Generally speaking, the accumulation of active ingredients in LH varies greatly due to the different ecological environments. Accordingly, the quality of herbs also varies, which makes it difficult to standardize commercial herbs and ensure their effectiveness for clinical use. Therefore, it is of great importance to study the constituents of LH from different habitats in Sichuan Province. This experiment focused on the compositional analysis of herbs from eight different habitats (Bazhong, Guangyuan, Yibin, Zigong, Deyang, Suining, Leshan and Nanchong) in Sichuan Basin. The PCA showed that the samples from the eight different habitats in Sichuan basin were clustered into one category and the distribution results were relatively ideal. According to the results of PLS-DA and VIP, four common differential chemical constituents were screened out from samples in Sichuan basin. The content of all four differential constituents in the samples of LH in Sichuan Bazhong was high and the overall level was good; the content of all four differential constituents in the sample of LH from Sichuan Yibin was low. However, the influence of different habitats on the quality of LH is still unclear in many aspects. Therefore, it is necessary to analyze the LH constituents of the Sichuan Plateau and to further develop a comparison of all samples from Sichuan. The results provide basic data for revealing the influence of the ecological environment on the synthesis and accumulation of metabolites of LH, as well as the quality formation mechanism of the herb.

## 4. Materials and Methods

### 4.1. Chemicals and Reagents

Threonine, glutamic acid, phenylalanine, caffeic acid, hyperoside, ferulic acid, vitexin, quercetin and rutin were supplied by the National Institutes for Food and Drug Control (Beijing, China). *p*-Coumaric acid, gallic acid, kaempferol 3-*O*-galactoside, isoschaftoside, schaftoside, isovitexin, astragalin, kaempferol 3-*O*-rutinoside and syringic acid were purchased from Chengdu Dexter Bio Co. (Chengdu, China). Protocatechuic acid, kaempferol and neochlorogenic acid were acquired from Shanghai yuanye Bio-Technology Co., Ltd. (Shanghai, China). Chlorogenic acid was received from Baoji Chenguang Biotechnology Co., Ltd. (Baoji, China). Isoquercetin was received from Jiangsu Yongjian Pharmaceutical Technology Co. (Taizhou, China). The purity of all compounds was above 98% as determined by high-performance liquid chromatography (HPLC). Methanol, formic acid and acetonitrile of HPLC grade were obtained from Merck (Darmstadt, Germany); ultrapure water was prepared using a Milli-Q water purification system (Millipore, Bedford, MA, USA).

### 4.2. Plant Materials

The samples of LH were collected in the field in Sichuan Province. The materials were identified by Professor Xunhong Liu (Department for Authentication of Chinese Medicines, School of Pharmacy, Nanjing University of Chinese Medicine, Nanjing, China) as the dried whole herb of *Lysimachia christinae* Hance. Voucher specimens were deposited in the laboratory of Chinese medicine identification, Nanjing University of Chinese Medicine. The source information of the LH samples is shown in Table 3.

### 4.3. UFLC-Triple TOF-MS/MS Analysis of LH

#### 4.3.1. Preparation of Standard and Sample Solutions

The above 23 standards were weighed in a 5 mL volumetric flask using a 1/1,000,000 electronic analytical balance (ME36S, Sydos, Germany), dissolved in methanol and prepared into the corresponding concentrations of standard solutions. Then, 50 μL of the above standard solutions was placed into a 10 mL volumetric flask and methanol was added to prepare a mixed standard solution. The concentration of the mixed standard solution was 5 μg/mL. All solutions were stored at 4 °C for further analysis.

The samples were crushed and passed through a No. 3 sieve and then the powder was dried to constant weight. Next, 0.5 g of dried powder was accurately weighed into a 50 mL centrifuge tube and ultrasonically extracted with 5 mL of 80% methanol for 60 min at room temperature. After extraction for a few minutes, the weight was made up with 80% methanol. The supernatant was taken and centrifuged at 13,000 rpm for 10 min before filtering through a 0.22 μm membrane (Jinteng laboratory equipment Co., Ltd., Tianjin, China) prior to UFLC-triple TOF-MS/MS analysis.

#### 4.3.2. UFLC-Triple TOF-MS/MS Conditions

A UFLC system (Shimadzu, Kyoto, Japan) was used for sample analysis. The separation was conducted by an Agilent ZORBAX SB-C18 column (4.6 mm × 250 mm, 5 μm) at 30 °C. The mobile phase consisted of methanol (A) and 0.1% formic acid in water (B) with the following gradient elution: 0–2 min (5–30% A), 2–25 min (30–55% A), 25–30 min (55–5% A) and 30–32 min (5–5% A). The column temperature was 30 ℃. The detection wavelength was 360 nm. The injection volume was 10 μL and the flow rate was 0.8 mL/min.

MS data were recorded using an AB Sciex Triple TOF^TM^ 5600 system-MS/MS (AB SCIEX, Framingham, MA, USA), equipped with an electrospray ionization (ESI) source. The optimized MS conditions in negative ion mode were set as follows: ion source temperature (TEM), 600 °C; flow rate of curtain gas (CUR), 40 psi; flow rate of nebulization gas (GS1) and flow rate of auxiliary gas (GS2), 60 psi; ion spray voltage floating (ISVF), 4500 V; collision energy, −10 V; declustering potential, −100 V. Data were acquired for each sample from 50 to 1500 Da. The data acquisition time was 32 min.

#### 4.3.3. Identification of Chemical Constituents

Generally, common characteristic peaks were identified by comparison with the reference compounds. If no standard reference was available for the compound, the characteristic peaks were inferred using Peakview 1.2 software (AB SCIEX, Framingham, MA, USA) and online resources, such as SciFinder (https://scifinder.cas.org/ (accessed on 13 May 2022)), HMDB (https://hmdb.ca/ (accessed on 20 May 2022)) and CNKI (https://kns.cnki.net/ (accessed on23 May 2022)) by comparing MS/MS fragment ions.

### 4.4. Analysis of the Differential Constituents in LH from Different Habitats

Markerview 1.2.1 software (Sciex AB, Framinghan, MA, USA) was used to perform peak matching, peak alignment and noise filtering for the raw mass spectrometry data. The results were further imported into SIMCA-P 13.0 software (Umetrics AB, Umea, Sweden). On the basis of the above qualitative results, PCA and PLS-DA were used to perform dimensionality reduction analysis on the data to obtain information about differences between groups. The characteristic peaks of the differential chemical components were screened according to the VIP (VIP > 1) and *t*-test (*p* < 0.05) results obtained from PLS-DA.

## 5. Conclusions

In this study, UFLC-triple TOF-MS/MS was used to analyze the components from eight different habitats in Sichuan basin. According to relevant mass spectrometry data, reference materials and the literature, 46 chemical constituents were identified. The fragmentation pathways of flavonoids, phenolic acids, tannins, amino acids, fatty acids and coumarins were preliminarily deduced by the fragmentation behavior of the major components. PCA, PLS-DA and *t*-tests were used to identify the different common chemical components of LH from different habitats and their different contents were compared at the same time. Finally, we found that LH from Sichuan Bazhong was best among the eight habitats. In conclusion, these results can help us to better understand the chemical constituents and componential differences of chemical constituents in LH from different habitats, as well as provide data for further exploring the functional material basis and clinical application of LH.

## Figures and Tables

**Figure 1 molecules-27-04600-f001:**
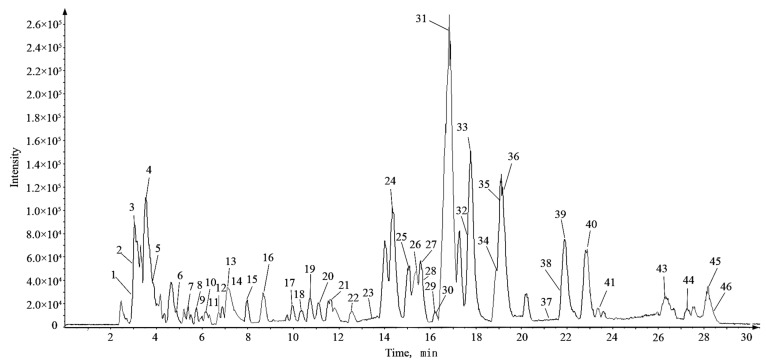
The base peak chromatogram (BPC) of Lysimachiae Herba from Sichuan Bazhong in negative ion mode.

**Figure 2 molecules-27-04600-f002:**
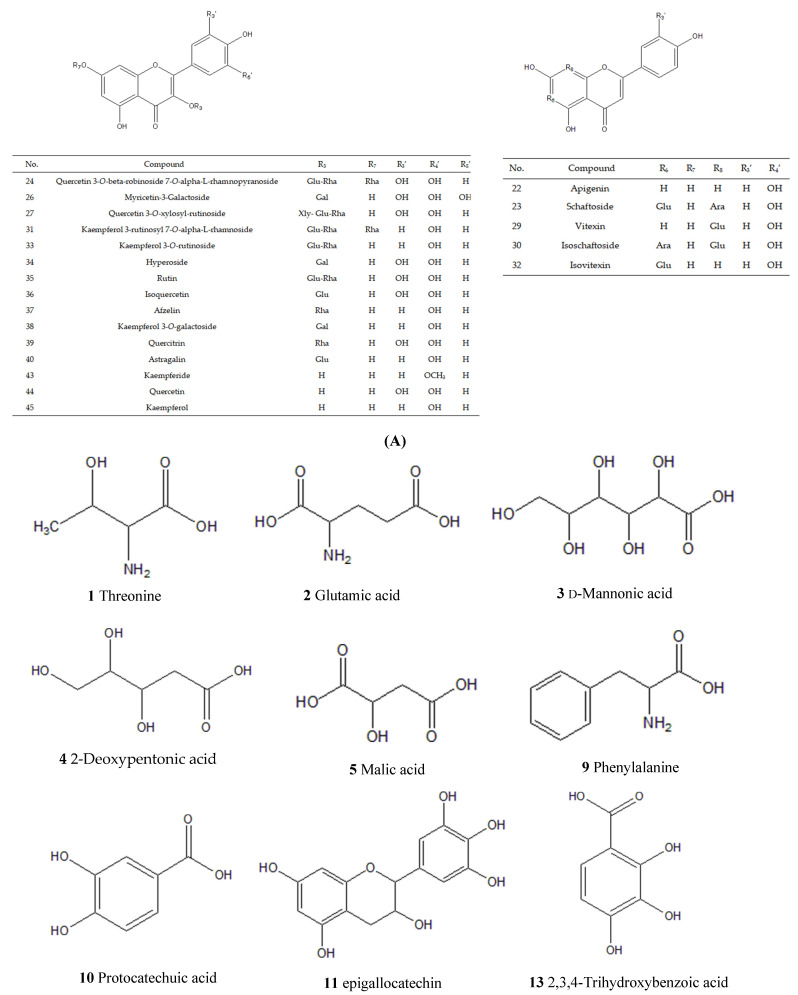
Chemical structures of constituents identified in Lysimachiae Herba: (**A**) general structures; (**B**) exact structures.

**Figure 3 molecules-27-04600-f003:**
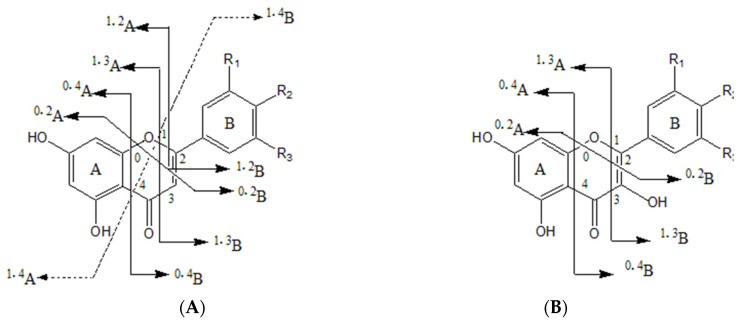
Schematic diagram of the fracture site of (**A**) flavonoid and (**B**) flavonol aglycone in negative ion mode.

**Figure 4 molecules-27-04600-f004:**
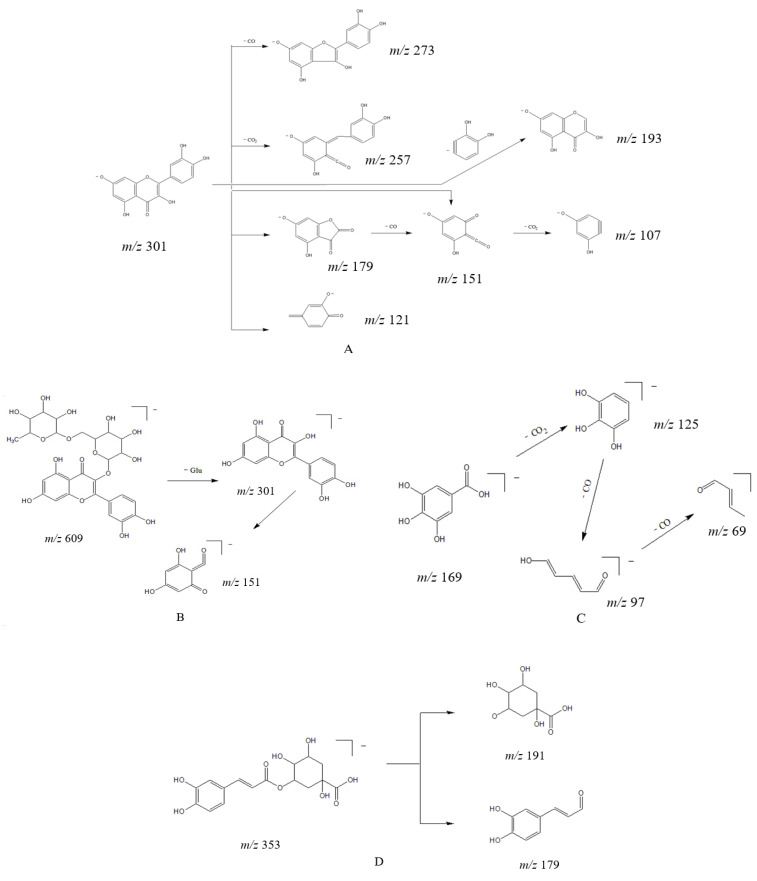
The speculated fragmentation pathways of (**A**) quercetin, (**B**) rutin, (**C**) gallic acid and (**D**) chlorogenic acid in Lysimachiae Herba.

**Figure 5 molecules-27-04600-f005:**
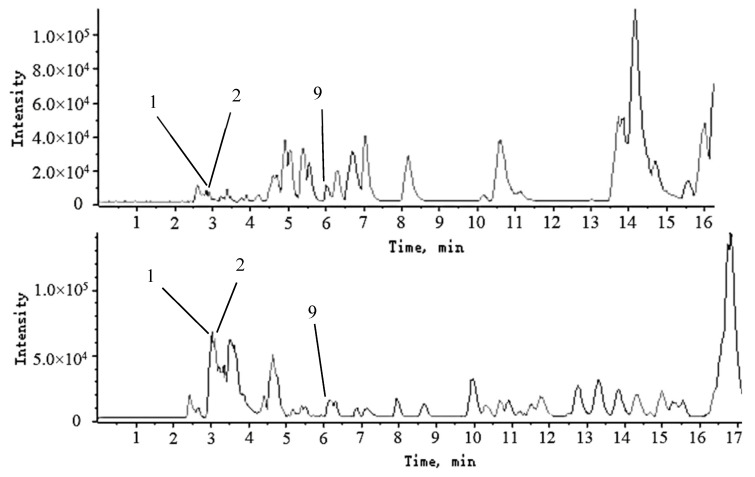
The base peak chromatogram (BPC) of part of the mixed standard solution (**a**) and the sample from Sichuan Deyang (**b**); (**1**) threonine, (**2**) glutamic acid, (**9**) phenylalanine.

**Figure 6 molecules-27-04600-f006:**
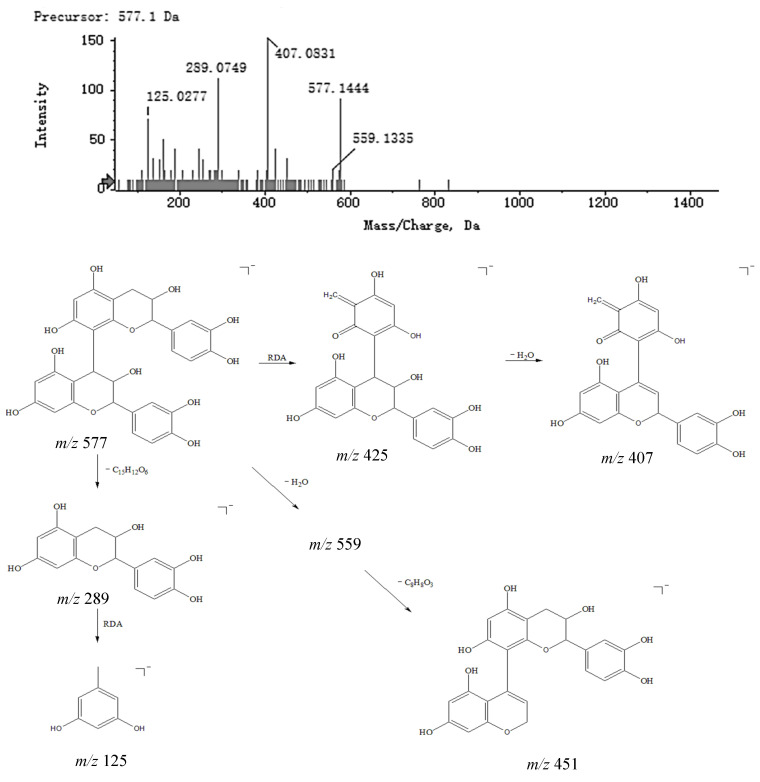
MS^2^ spectra of procyanidin B1 and speculated fragmentation pathways.

**Figure 7 molecules-27-04600-f007:**
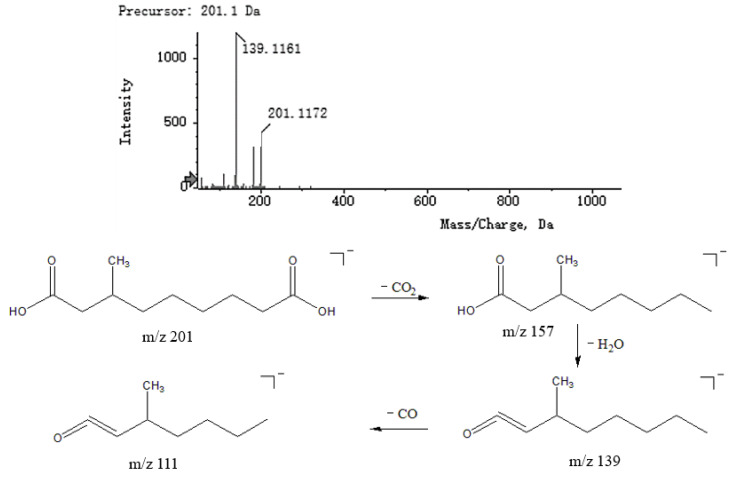
MS^2^ spectra of 3-methylnonanoic acid and speculated fragmentation pathways.

**Figure 8 molecules-27-04600-f008:**
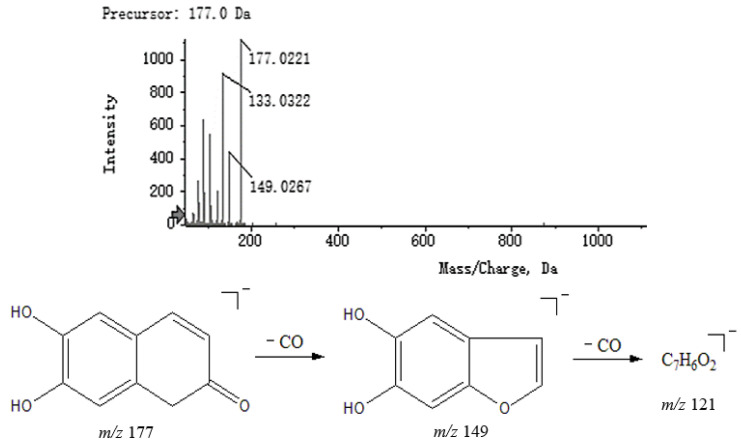
MS^2^ spectrum of 6,7-dihydroxycoumarin and speculated fragmentation pathways.

**Figure 9 molecules-27-04600-f009:**
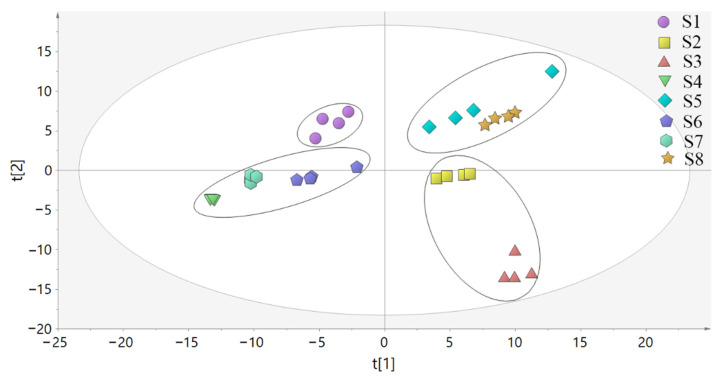
PCA score plot of LH samples from different habitats.

**Figure 10 molecules-27-04600-f010:**
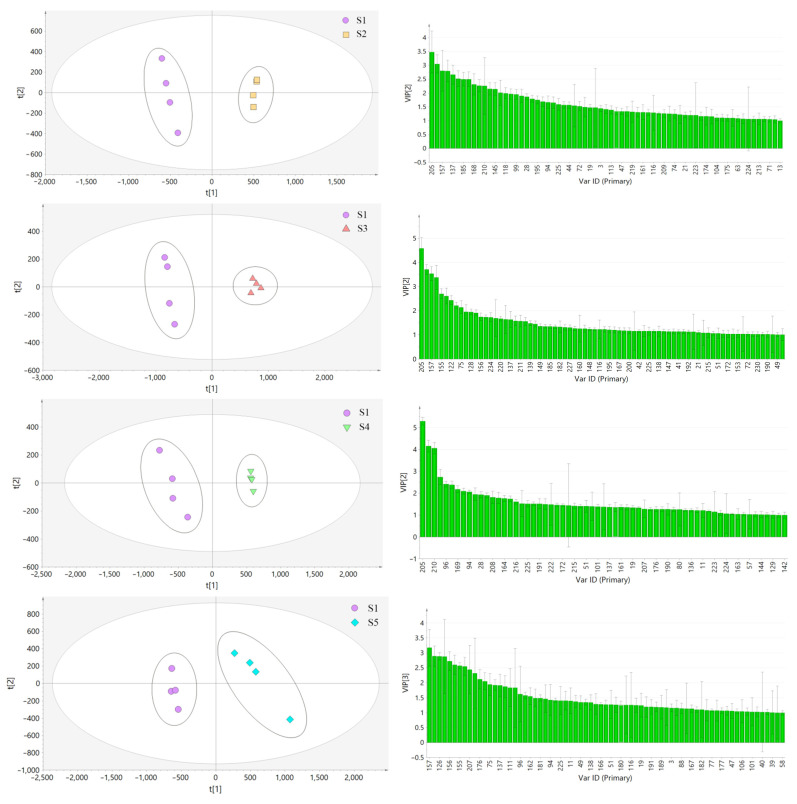
PLS-DA score plot (**a**) and VIP score plot (**b**) of LH samples from different habitats.

**Figure 11 molecules-27-04600-f011:**
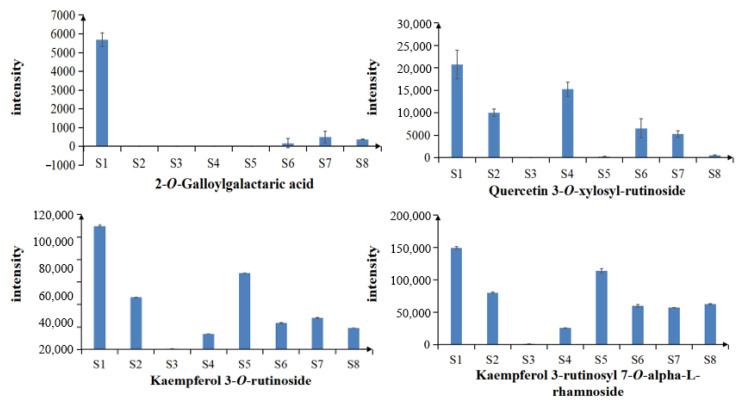
Relative contents of the common differential chemical constituents.

**Table 1 molecules-27-04600-t001:** Identification of 46 constituents in Lysimachiae Herba by UFLC-triple TOF-MS/MS.

No.	tR min	Molecular formula	[M − H]^− 1^	MS^2^	Error (ppm)	Compound	S1	S2	S3	S4	S5	S6	S7	S8	Ref.
1	2.90	C_4_H_9_NO_3_	118.0522	74.028 [M − H − CO_2_]^−^	4.90	Threonine ^1^	+	+	+	+	+	+	−	−	[9]
2	2.92	C_5_H_9_NO_4_	146.0466	56.0548 [M − H − CO_2_ − CO − H_2_O]^−^, 84.0464 [M − H − CO_2_ − H_2_O]^−^	4.60	Glutamic acid ^1^	+	+	+	+	+	+	+	+	[10]
3	3.03	C6H12O7	195.0543	59.0177, 75.0119, 129.0218, 177.0436 [M − H − H_2_O]^−^	0.36	d-Mannonic acid	+	+	+	+	+	+	+	+	[11]
4	3.47	C5H10O5	149.0486	73.0329 [M − H − H_2_O − CO_2_ − CH_2_]^−^, 89.0273 [M − H − H_2_O − C_3_H_6_]^−^, 105.0218 [M − H − CO_2_]^−^	4.36	2-Deoxypentonic acid	+	+	+	+	+	+	+	+	[12]
5	3.76	C4H6O5	133.0176	71.0165 [M − H − H_2_O − CO_2_]^−^, 115.0060 [M − H − H_2_O]^−^	4.98	Malic acid	+	−	+	+	+	+	+	+	[13]
6	4.87	C30H26O14	609.1355	177.0235 [M − H − C_15_H_12_O_6_ − C_6_H_8_O_3_]^−^, 305.0714 [M − H − C_15_H_12_O_6_]^−^, 423.0776 [M − H − C_8_H_10_O_5_]^−^	−1.62	Prodelphinidin B1	+	−	−	−	+	−	−	−	[12]
7	5.35	C7H6O5	169.0143	69.0390 [M − H − CO_2_ − 2CO]^−^, 97.0322 [M − H − CO_2_ − CO]^−^, 107.0141 [M − H − CO_2_ − H_2_O]^−^, 125.0248 [M − H − CO_2_]^−^	3.25	Gallic acid ^1^	+	+	+	+	+	+	+	+	[14]
8	5.74	C30H26O12	577.1444	125.0277 [^1,4^A]^−^, 179.0721 [M − H − C_15_H_12_O_6_ − C_6_H_5_O_2_]^−^, 245.0743 [M − H − C_15_H_12_O_6_ − CO_2_]^−^, 289.0749 [M − H − C_15_H_12_O_6_]^−^, 407.0931 [M − H − C_8_H_8_O_3_ − H_2_O]^−^, 425.0829 [M − H − C_8_H_8_O_3_]^−^, 451.0986 [M − H − H_2_O − C_6_H_5_O_2_]^−^, 559.1335 [M − H − H_2_O]^−^	−1.65	Procyanidin B1	+	+	+	+	−	+	+	+	[15]
9	6.04	C_9_H_11_NO_2_	164.0723	103.0595 [M − H − NH_3_]^−^, 147.0471 [M − H − NH_3_ − CO_2_]^−^	1.71	Phenylalanine ^1^	+	+	+	+	+	+	+	+	[16]
10	6.15	C7H6O4	153.0199	81.0361 [M − H − CO − CO_2_]^−^, 91.0197 [M − H − CO_2_ − H_2_O]^−^, 109.0298 [M − H − CO_2_]^−^, 125.0323 [M − H − CO]^−^	1.70	Protocatechuic acid ^1^	+	+	+	+	+	+	+	+	[17]
11	6.66	C15H14O7	305.0661	125.0275 [^1,4^A]^−^, 137.0266 [^1,3^A]^−^, 167.0339 [^1,2^A]^−^	−0.95	Epigallocatechin	+	+	+	+	+	+	+	+	[18]
12	6.70	C16H18O9	353.0868	135.0478 [M − H − C_7_H_10_O_5_ − CO_2_]^−^, 179.0391 [M − H − C_7_H_10_O_5_]^−^, 191.0596 [M − H − caffeoyl]^−^	−2.86	Neochlorogenic acid ^1^	+	+	+	+	+	+	+	+	[19]
13	7.17	C7H6O5	169.0146	83.0170 [M − H − CO_2_ − 2H_2_O]^−^, 125.0271 [M − H − CO_2_]^−^, 151.0064 [M − H − H_2_O]^−^	2.07	2,3,4-Trihydroxybenzoic acid	+	+	+	+	+	+	+	+	[20]
14	7.24	C13H14O12	361.0108	125.0278, 151.0065, 169.0177	−1.11	2-O-Galloylgalactaric acid	+	+	+	+	+	+	+	+	[11]
15	8.22	C16H18O9	353.0872	85.0311, 161.0231 [M − H − C_7_H_10_O_5_ − H_2_O]^−^, 179.0391 [M − H − C_7_H_10_O_5_]^−^, 191.0596 [M − H − caffeoyl]^−^	−1.73	Chlorogenic acid ^1^	+	+	+	+	+	+	+	+	[21]
16	8.67	C27H32O15	595.1673	269.0876 [M − H − RG − H_2_O]^−^, 287.0995 [M − H − RG]^−^	0.77	Eriocitrin	+	+	+	+	+	+	+	+	[22]
17	9.97	C21H22O11	449.1063	269.0498 [M − H − Glc − H_2_O]^−^, 287.0599 [M − H − Glc]^−^	−4.16	Marein	+	+	−	+	+	+	+	+	[23]
18	10.29	C9H6O4	177.0221	121.0327 [M − H − 2CO]^−^, 149.0269 [M − H − CO]^−^	3.78	6,7-Dihydroxycoumarin	+	+	+	+	+	+	+	+	[12]
19	10.49	C9H8O4	179.0351	89.0413 [M − H − CO_2_ − CO − H_2_O]^−^, 109.0440 [M − H − CO_2_ − CO]^−^, 135.0449 [M − H − CO_2_]^−^	0.22	Caffeic acid ^1^	+	+	+	+	+	+	+	+	[21]
20	11.37	C9H10O5	197.0460	123.0073 [M − H − C_3_H_6_O_2_]^−^, 167.0022 [M − H − CH_2_O]^−^	2.28	Syringic acid ^1^	+	+	+	+	+	+	+	+	[24]
21	11.60	C13H12O8	295.0464	115.0060, 133.0167	1.56	Caffeoylmalic acid	+	+	+	+	+	−	−	+	[11]
22	12.65	C15H10O5	269.0564	107.0192 [^1,3^A − CO_2_]^−^, 117.0345 [M − H − C_7_H_4_O_4_]^−^, 151.0072 [^1,3^A]^−^, 225.0652 [M − H − CO_2_]^−^	4.70	Apigenin	+	+	+	−	−	+	+	−	[21]
23	13.52	C26H28O14	563.1391	383.0761 [M − H − 2C_3_H_6_O_3_]^−^, 443.0970 [M − H − C_4_H_8_O_4_]^−^, 473.1079 [M − H − C_3_H_6_O_3_]^−^	−2.70	Schaftoside ^1^	+	+	+	+	+	+	+	+	[25]
24	14.39	C33H40O20	755.2041	255.0334, 271.0286, 301.0402 [M − H − RG − Rha]^−^	0.11	Quercetin 3-*O*-beta-robinoside 7-*O*-alpha-l-rhamnopyranoside	+	+	+	+	+	+	+	+	[11]
25	15.05	C9H8O3	163.0405	93.0356 [M − H − C_3_H_2_O_2_]^−^, 119.0508 [M − H − CO_2_]^−^	2.64	*p*-Coumaric acid ^1^	+	+	+	+	+	+	+	+	[26]
26	15.23	C21H20O13	479.0825	271.0294, 287.0249, 317.0366 [M − H − Gal]^−^	−1.27	Myricetin-3-galactoside	+	+	+	+	+	+	+	+	[12]
27	15.59	C32H38O20	741.1854	179.0022 [M − H − RG − Xyl − H_2_O]^−^, 301.0404 [M − H − RG − Xyl]^−^	−1.82	Quercetin 3-*O*-xylosyl-rutinoside	+	+	+	+	+	+	+	+	[11]
28	15.8	C10H10O4	193.0507	133.0319 [M − H − C_2_H_4_O_2_]^−^	0.71	Ferulic acid ^1^	+	+	+	+	+	+	+	+	[27]
29	16.12	C21H20O10	431.0986	311.0594 [M − H − C_4_H_8_O_4_]^−^, 341.1187 [M − H − C_3_H_6_O_3_]^−^	0.53	Vitexin ^1^	+	+	−	+	+	+	+	+	[28]
30	16.18	C26H28O14	563.1393	383.0765 [M − H − 2C_3_H_6_O_3_]^−^, 443.0971 [M − H − C_4_H_8_O_4_]^−^	−3.60	Isoschaftoside ^1^	+	+	+	−	+	+	+	+	[21]
31	16.76	C33H40O19	739.2051	227.0382, 255.0329, 285.0431 [M − H − RG − Rha]^−^	−3.66	Kaempferol 3-rutinosyl 7-*O*-alpha-l-rhamnoside	+	+	+	+	+	+	+	+	[12]
32	17.56	C21H20O10	431.0982	269.0495 [M − H − Rha]^−^, 311.0590 [M − H − C_4_H_8_O_4_]^−^, 341.0688 [M − H − C_3_H_6_O_3_]^−^	0.39	Isovitexin ^1^	+	+	−	+	+	+	+	+	[25]
33	17.97	C27H30O15	593.1495	285.0393 [M − H − RG]^−^	−2.85	Kaempferol 3-*O*-rutinoside ^1^	+	+	+	+	+	+	+	+	[29]
34	18.65	C21H20O12	463.0872	301.0408 [M − H − Gal]^−^	−2.16	Hyperoside ^1^	+	+	+	+	+	+	+	+	[21]
35	18.82	C27H30O16	609.1461	151.0035 [^1,3^A]^−^, 301.0353 [M − H − RG]^−^	−0.02	Rutin ^1^	+	+	+	+	+	+	+	+	[21]
36	18.94	C21H20O12	463.0887	301.0498 [M − H − Glc]^−^	−2.26	Isoquercetin ^1^	+	+	+	+	+	+	+	+	[21]
37	21.22	C21H20O10	431.0963	151.0073 [^1,3^A]^−^, 269.0489 [M − H − Rha]^−^, 413.2366 [M − H − H_2_O]^−^	−4.80	Afzelin	+	+	+	+	+	+	+	+	[30]
38	21.74	C21H20O11	447.0902	151.0070 [^1,3^A]^−^, 285.0447 [M − H − Gal]^−^	−3.17	Kaempferol 3-*O*-galactoside ^1^	+	+	+	+	+	+	+	+	[31]
39	21.84	C21H20O11	447.0901	301.0402 [M − H − Rha]^−^	−3.65	Quercitrin	+	+	+	+	+	+	+	+	[32]
40	22.76	C21H20O11	447.0894	151.0065 [^1,3^A]^−^, 285.0450 [M − H − Glc]^−^	−3.80	Astragalin ^1^	+	+	+	+	+	+	+	+	[33]
41	23.33	C10H18O4	201.1172	111.0852 [M − H − CO_2_ − H_2_O − CO]^−^, 139.1161 [M − H − CO_2_ − H_2_O]^−^, 183.1056 [M − H − H_2_O]^−^	4.17	3-Methylazelaic acid	+	+	+	+	+	+	+	+	[11]
42	26.35	C15H10O6	285.0404	227.0409 [M − H − 2CHO]^−^	0.10	Luteolin	+	+	+	+	+	−	−	+	[28]
43	26.38	C16H12O6	299.0561	284.0307 [M − H − CH_3_]^−^	−0.21	Kaempferide	+	−	−	+	−	+	+	−	[21]
44	27.21	C15H10O7	301.0348	107.0172 [^0,4^A]^−^, 121.0483 [M − H − ^1,3^A]^−^, 151.0034 [^1,3^A]^−^, 179.0341 [^1,2^A]^−^, 193.0287 [M − H − B ring]^−^, 257.0401 [M − H − CO_2_]^−^, 273.0393 [M − H − CO]^−^	−1.93	Quercetin ^1^	+	+	+	+	+	+	+	+	[21]
45	28.34	C15H10O6	285.0401	133.0295 [^1,3^A − H_2_O]^−^, 151.0044 [^1,3^A]^−^, 162.8200 [^0,2^A]^−^,	−1.22	Kaempferol ^1^	+	+	+	+	+	+	+	+	[21]
46	28.43	C15H12O5	271.0610	119.0529 [M − H − C_7_H_4_O_4_]^−^, 151.0073 [^1,3^A]^−^	0.61	Naringenin	+	+	−	−	+	−	+	−	[30]

Note: ^1^ Comparison with reference standards.

**Table 2 molecules-27-04600-t002:** PLS-DA results of LH samples from different habitats in Sichuan Basin.

Samples	Model Verification Results			Permutation Results		Number of Characteristic Peaks with VIP > 1
	*R*^2^*X* (cum)	*R*^2^*Y* (cum)	*Q*^2^ (cum)	*R* ^2^	*Q* ^2^	
S1, S2	0.888	0.996	0.990	0.460	−0.225	67
S1, S3	0.905	0.996	0.990	0.143	−0.273	73
S1, S4	0.871	0.969	0.960	0.099	−0.243	63
S1, S5	0.794	0.892	0.835	0.453	−0.082	71
S1, S6	0.680	0.930	0.903	0.239	−0.215	64
S1, S7	0.789	0.995	0.922	0.300	−0.230	56
S1, S8	0.842	0.990	0.981	0.312	−0.244	43

**Table 3 molecules-27-04600-t003:** Information of Lysimachiae Herba samples from eight different habitats.

No.	Sichuan Habitats	Longitude and Latitude
S1	Yuxi Town, Enyang District, Bazhong City	N: 31°49′00.16″ E: 106°29′12.02″
S2	Puan Town, Jiange County, Guangyuan City	N: 32°01′59.82″ E: 105°28′12.96″
S3	Zhuhai Town, Changning County, Yibin City	N: 28°29′56.84″ E: 104°55′57.69″
S4	Dongjia Town, Rong County, Zigong City	N: 29°18′15.09″ E: 104°10′48.66″
S5	Taian Town, Zhongjiang County, Deyang City	N: 30°37′33.17″ E: 104°57′17.13″
S6	Qunli Town, Pengxi County, Suining City	N: 30°22′59.82″ E: 105°58′30.98″
S7	Mata Town, Jingyan County, Leshan City	N: 29°31′20.89″ E: 103°59′48.30″
S8	Daqiao Town, Nanbu County, Nanchong City	N: 31°21′34.50″ E: 105°46′57.72″

## Data Availability

The data presented in this study are available within the article.

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
