# Peer review of "Qualitative Analysis and Componential Differences of Chemical Constituents in Lysimachiae Herba from Different Habitats (Sichuan Basin) by UFLC-Triple TOF-MS/MS"

_molecules, 2022, doi:10.3390/molecules27144600_

Round 1

Reviewer 1 Report

I found the Manuscipt title and abstract quite interesting and highly relevant to students working on identification and characterization of chemical constituents from traditional medicines.

To strengthen the manuscript following major revisions are required:

1) How the authors presumed Componds 1,2 and 9 as Threonine, glutamic acid and phenylanalnine in 2.3.3. Identification of Amino Acids [Line no 227, Page number 9 of 18]. Authors to show the comparison of chromatograms of all 3 standards with that of extract.

2) Preassumption of Componds 6 and 8 as condensed ellagitannins? 2.3.4. Identification of Tannins [Line no 231, Page number 9 of 18]. Show MS/MS pattern and also Strengthen it with the Schematic Diagram.

 3) Proof to show Compounds 3, 4, 5 and 41 as Fatty Acids?  2.3.5. Identification of Fatty acids [Line no 245, Page number 9 of 18]. Show the MS/MS pattern? Stregthen by Schematic diagram of fracture site. Similarly do it for Coumarins. 

Minor Revisions and clarifications Required

1) What was the weight of all 23 standards? [Line no 371, 372, Page number 15 of 18]. Mentions concentrations of Standard solutions. What was the concentration of mixed standard solution? (Mention details of weighing balance used as well) 

1) Please explain Sentence line no 51 to 53, Page number 2 of 18 Meaning not clear.

2) What the author are referring to as higha nd low in line no 56 and 57, Page number 2 of 18? Kindly explain?

Typographical errors

1) Correction in numbering of Section Identifiaction of Coumarins from 2.3.5 to 2.3.6 Line no 250, Page number 9 of 18.

 2) Space between 60 and 10 min in line number 377 and 379, Page numer 15 of 18. 

Author Response

I found the Manuscipt title and abstract quite interesting and highly relevant to students working on identification and characterization of chemical constituents from traditional medicines.

Reply: First of all, thank you very much for your recognition and encouragement of our work. We sincerely thank you for your enthusiastic work.

Point 1 How the authors presumed Componds 1,2 and 9 as Threonine, glutamic acid and phenylanalnine in 2.3.3. Identification of Amino Acids [Line no 227, Page number 9 of 18]. Authors to show the comparison of chromatograms of all 3 standards with that of extract.

Reply 1 Thanks for your professional comments. 

  • The identification process of 3 amino acids now presented in Line 227–234, Page number 12 of 21. The addition in the paper is as follows: “The 3 amino acids are identified mainly by comparison with the standards, the process is as follows: firstly, we use Peakview to find the retention times of the corresponding amino acids in the mixed standard solution; secondly, we compare the m/zof substances with similar retention times (within 0.5 min) in the samples and those with m/z errors of more than 5 ppm are removed; then the MS/MS patterns of the eligible amino acids are consistent with the references and relevant web queries. In the end, compounds 1, 2, and 9 are presumed to be threonine, glutamic acid, and phenylalanine, respectively.” 
  • The addition is also shown below. The comparison of the chromatograms of the 3 amino acid standards with those of the extract has been added to the article in Line 236, Page number 12 of 21. The addition is also shown below (Figure 1).

Figure 1 The base peak chromatogram (BPC) of part of mixed standard solution (a) and sample from Sichuan Deyang (b) 1. Threonine, 2. Glutamic acid, 9. Phenylalanine.

Point 2 Preassumption of Componds 6 and 8 as condensed ellagitannins? 2.3.4. Identification of Tannins [Line no 231, Page number 9 of 18]. Show MS/MS pattern and also Strengthen it with the Schematic Diagram.

Reply 2 Thanks for your professional comments. 

  • It is reasonable to speculate that compounds 6 and 8 are condensed ellagitannins. The identification process now presented in Line 241-247, Page number 12 of 21. The addition is as follows: “The process of identification is as follows: first, the results of the analysis are imported into Peakview; then compounds that meet the mass error of less than 5 ppm, have the correct isotopic distribution and contain secondary fragments are identified as targets. Combining features of Peakview such as Formula Finder, matching the Mass spectrometry data of each chromatographic peak in the database (SciFinder and HMDB), and the cleavage law of each peak, eventually compounds 6 and 8 are identified as prodelphinidin B1 and procyanidin B1.”
  • We have put the MS2spectra and speculated fragmentation pathways of compound 8 (procyanidin B1) in the article in Line 255, Page number 13 of 21. The addition along with MS2 spectra of compound 6 (prodelphinidin B1) is shown below (Figure 2 and 3).

Figure 2. MS2 spectra of Procyanidin B1 and speculated fragmentation pathways.

Figure 3. MS2 spectra of compound 6 (prodelphinidin B1).

Point 3 Proof to show Compounds 3, 4, 5 and 41 as Fatty Acids?  2.3.5. Identification of Fatty acids [Line no 245, Page number 9 of 18]. Show the MS/MS pattern? Stregthen by Schematic diagram of fracture site. Similarly do it for Coumarins. 

Reply 3 Thanks for your professional comments. 

  • The proof is the corresponding literature for fatty acids and the results of software analysis. Taking compound 4 as an example, the molecular ion peak at a retention time of 3.47 min was obtained at m/z 149.0486[M−H]and the secondary mass spectra showed peaks at m/z 0329, 89.0273, and 105.0218. According to the database (HMDB), the molecular formula is C5H10O5, and the compound is presumed to be 2-Deoxypentonic acid by comparison with the fragments consulted in the literature.
  • The MS/MS patterns of compounds 3, 4, and 5 are shown as follow (Figure 4, 5 and 6). The schematic diagram of compound 41 along with the MS/MS patterns are added in the article in Line 268, Page number 14 of 21 and is also shown below (Figure 7).

Figure 4. MS/MS patterns of Compound 3 (D-Mannonic acid).

Figure 5. MS/MS patterns of Compound 4 (2-Deoxypentonic acid).

Figure 6. MS/MS patterns of Compound 5 (Malic acid).

Figure 7. MS2 spectra of 3-Methylnonanoic acid and speculated fragmentation pathways.

  • The molecular ion peak at a retention time of 10.29 min was obtained at m/z0221 [M−H] and the secondary mass spectra showed peaks at m/z 121.0327 and m/z 149.0269. According to the database (HMDB), the molecular formula is C9H6O4, and the compound is presumed to be 6,7-Dihydroxycoumarin by comparsion with the fragments consulted in the literature. And 6, 7-dihydroxycoumarin is classified as coumarin. The MS2 spectra of 6,7-Dihydroxycoumarin and speculated fragmentation pathways are shown as follows (Figure 8).

Figure 8. MS2 spectra of 6,7-Dihydroxycoumarin and speculated fragmentation pathways.

Point 4 What was the weight of all 23 standards? [Line no 371, 372, Page number 15 of 18]. Mentions concentrations of Standard solutions. What was the concentration of mixed standard solution? (Mention details of weighing balance used as well) 

Reply 4 Thanks for your professional comments. 

  • The weights of 23 standards are shown in Table 1.
  • The concentration of mixed standard solution is added in Line no 369, 370, Page number 19 of 21 and added as: “The concentration of the mixed standard solution was 5 μg/mL.”
  • The detail of weighing balance have been added in the article in Line no 365, 366, Page number 19 of 21. The addition is as follows: “The above 23 standards were weighed in a 5 mL volumetric flask by using 1/1,000,000 electronic analytical balance (ME36S, Sydos, Germany), dissolved in methanol and prepared into the corresponding concentrations of standard solutions.”

Table 1. The weights of 23 standards

No.

Standard

Weight (mg)

1

Threonine

5.80

2

Glutamic acid

4.80

3

Gallic acid

4.85

4

Phenylalanine

5.82

5

Protocatechuic acid

5.18

6

Neochlorogenic acid

4.97

7

Caffeic acid

5.26

8

Syringic acid

5.06

9

Schaftoside

5.17

10

P-coumaric acid

5.04

11

Ferulic acid

4.99

12

Vitexin

5.59

13

Isoschaftoside

5.09

14

Isovitexin

4.97

15

Kaempferol 3-O-rutinoside

4.96

16

Hyperoside

4.68

17

Rutin

5.10

18

Isoquercetin

5.47

19

Kaempferol 3-O-galactoside

5.02

20

Astragalin

5.14

21

Quercetin

5.16

22

Kaempferol

4.82

23

Naringenin

5.07

Point 5 Please explain Sentence line no 51 to 53, Page number 2 of 18. Meaning not clear.

Reply 5 Thanks for your sincere reminder. This sentence was to explain the meaning of “genuine Chinese herbs”. Generally speaking, genuine Chinese herbs are produced in a particular area. Because of the special ecological environment of the particular area, the quality of the herbs is excellent and clinically certified. We have revised this sentence presented in line no 50 to 51, Page number 2 of 21 to make it clearer and easier to understand, revised as follows: “It is commonly believed that genuine Chinese herbs refer to medicinal materials produced in a specific area of natural conditions.”

Point 6 What the author are referring to as high and low in line no 56 and 57, Page number 2 of 18? Kindly explain?

Reply 6 Thanks for your sincere question. “High and low” here refers to the elevation of the land. The eastern part of Sichuan is the Sichuan Basin and mountains which locate on the margin of the basin; The western part is the Sichuan plateau and mountains and the highland of southwest Sichuan; The highest point is Gongga Mountain, rising to 7,556 meters above sea level; The lowest point is in the east of Sichuan, only 184 meters above sea level and 7,300 meters lower than Gongga Mountain. So, the terrain is high in the west and low in the east, and tilts from northwest to southeast. We have revised this sentence presented in line no 55 to 57, Page number 2 of 21 to make it clearer and easier to understand, revised as follows: “Sichuan Province is located inland in southwest China and the upper reaches of the Yangtze River, and its terrain is characterized by high in the west and low in the east.”

Point 7 Typographical errors

  • Correction in numbering of Section Identifiaction of Coumarins from 2.3.5 to 2.3.6 Line no 250, Page number 9 of 18.

2) Space between 60 and 10 min in line number 377 and 379, Page numer 15 of 18. 

Reply 7 Thanks for your careful observation and we are very sorry for our negligence.

  • The revised part was presented in the Line no 269, Page number 14 of 21, revised as follows: “2.3.6. Identification of Coumarins”.
  • The revised part was presented in the Line no 372 and 378, Page number 19 of 21, revised as follows: “Accurately 0.5 g of dried powder was weighed into a 50 mL centrifuge tube and ultrasonically extracted with 5 mL of 80% methanol for 60 min at room temperature. After the extraction was placed for a few minutes, made up the weight with 80% methanol. The supernatant was taken and centrifuged at 13000 rpm/min for 10 min and filtered through 0.22 μm membrane (Jinteng laboratory equipment Co., Ltd., Tianjin, China) prior to injection of UFLC-Triple TOF-MS/MS analysis.”

Once again, thank you very much for your comments and advice.

Reviewer 2 Report

This is an interesting paper on plant metabolomics by LC-MS/MS, describing a Qualitative Analysis and Componential Differences of Chemical Constituents in Lysimachiae Herba from Different Habitats (Sichuan Basin).

There is no major objection from my side on the scientific work of this study, which, indeed, presents interesting findings. However, comparison with previous papers is missing. This is critical, as the novelty of the paper can not be shown (both in Introduction and in Results & Discussion).

Why did authors skip detection in positive ion mode, as well? Indeed, negative ion mode detection is favoured in most cases, but for instance, amino acids are determined mainly in positive ion mode.

Author Response

Response to Reviewer 2 Comments

This is an interesting paper on plant metabolomics by LC-MS/MS, describing a Qualitative Analysis and Componential Differences of Chemical Constituents in Lysimachiae Herba from Different Habitats (Sichuan Basin).

Reply: First of all, thank you very much for your recognition and encouragement of our work. We sincerely thank you for your enthusiastic work.

Point 1 However, comparison with previous papers is missing. This is critical, as the novelty of the paper can not be shown (both in Introduction and in Results & Discussion).

Reply 1 Thanks for your professional advice. We have written some comparisons in the introduction, results, and discussion, taking the introduction as an example: first, the contents of kaempferol and quercetin in LH are considered as quality standards in the Chinese Pharmacopoeia (2020 version), which is one-sided, so it is important to find a comprehensive quality control of LH [Line no 46-49, Page number 2 of 21]; second, the authentic area of LH is not clear in many references, but the ecological environment may vary greatly in different areas of the province [Line no 53-59, Page number 2 of 21]. Therefore, samples from 8 habitats in Sichuan Basin were seleted to study the differences in the chemical composition; third, UFLC-Triple TOF-MS/MS technique is used for the analysis of the main chemical constituent in LH, which has the characteristics of high sensitivity, high selectivity, accurate and reliable determination results, and is more suitable for the separation and analysis [Line no 64-67, Page number 2 of 21]; Compared with previous papers, these results will help us to better understand the chemical constituents and componential differences of chemical constituents in LH from different habitats, and provide data for further exploring the functional material basis and clinical application of LH. 

Point 2 Why did authors skip detection in positive ion mode, as well? Indeed, negative ion mode detection is favoured in most cases, but for instance, amino acids are determined mainly in positive ion mode.

Reply 2 Thanks for your kind advice. We detected samples in negative ion mode for the following reasons. Firstly, flavonoids and phenolic acids are the main active components of LH according to previous literature. Secondly, the components detected in positive ion mode can be detected in negative ion mode. At the same time, flavonoids and phenolic acids are also more sensitive in the negative ion mode. In conclusion, we choose the negative ion mode to detect the samples.

Figure 9. The base peak chromatogram (BPC) of Lysimachia Herba from Sichuan bazhong in the negative (a) and positive (b) ion mode

Once again, thank you very much for your comments and suggestions.

Reviewer 3 Report

The manuscript of Zhou et al. is an interesting experiment showing the chemical profiling of Lysimachiae herba from different habitats. My suggestions relate mainly to the mass identification and presentation of results.

Table 1. In “No.” remove “*”. Change “tR/min” to “tR min”. In “Molecular formula” expands the column. Change “MS1 (m/z)” to “[M-H]-1” and remove all “[M-H]-1“ below. In “MS2“ remove “m/z”. Change “Error/ppm” for “Error (ppm)”. In “Compound”, each substance linked as standard should be defined as the number 1, writing in “Note”, below, 1 as “compared with reference standards”. In “S1” make likewise of 1 modifying for 2 and in “Note” describe each habitat. Change “Reference” for “Ref.”. To identification of compounds is acceptable only ppm error of until 5 ppm in module. Remake the values for compounds 5, 17, 31, and 38. Remove the (2) in “Notes”. In compound name, try to expand the column or use them - to complete the name, for example to the compounds 4, 13, 14, 18, 24, 26.

Line 151. What is the standard? Kaempferol?

Line 269. After “the samples of LH from 8 habitats” show the legend of each habitat “S” (Bazhong, Guangyuan, Yibin, Zigong, Deyang, Suining, Leshan, and Nanchong), for example, S1, xx; S2, xx; S3, xxx and remove the habitat name in other text.

Figure 5. Please circle only the habitat clusters, removing the cycle in each substance. Centralize the figure.

The PCA and PLS models need to be improved. Please, try to use S-plot and VIP in the V graph to better explain the values.

Author Response

Response to Reviewer 3 Comments

The manuscript of Zhou et al. is an interesting experiment showing the chemical profiling of Lysimachiae herba from different habitats. My suggestions relate mainly to the mass identification and presentation of results.

Reply: First of all, thank you very much for your recognition and encouragement of our work. We sincerely thank you for your enthusiastic work.

Point 1 Table 1. In “No.” remove “*”. Change “tR/min” to “tR min”. In “Molecular formula” expands the column. Change “MS1 (m/z)” to “[M-H]-1” and remove all “[M-H]-1“ below. In “MS2“ remove “m/z”. Change “Error/ppm” for “Error (ppm)”. In “Compound”, each substance linked as standard should be defined as the number 1, writing in “Note”, below, 1 as “compared with reference standards”. In “S1” make likewise of 1 modifying for 2 and in “Note” describe each habitat. Change “Reference” for “Ref.”. To identification of compounds is acceptable only ppm error of until 5 ppm in module. Remake the values for compounds 5, 17, 31, and 38. Remove the (2) in “Notes”. In compound name, try to expand the column or use them - to complete the name, for example to the compounds 4, 13, 14, 18, 24, 26.

Reply 1 Thanks for your kind advice and we are very sorry for our negligence. We have followed your modification suggestion. And the detailed information of S1 — S8 is described in Table 3 in the Line no 362, Page number 18 of 21. Table 1 is now presented in line 120, and the revised table is as follows.

Table 1. Identification of 46 constituents in Lysimachiae Herba by UFLC-Triple TOF-MS/MS.

 No.

tR

min

Molecular formula

[M−H]1

MS2

Error

(ppm)

Compound

S1

S2

S3

S4

S5

S6

S7

S8

Ref.

1

2.90

C4H9NO3

118.0522

74.028[M−H−CO2]

4.90

Threonine1

+

+

+

+

+

+

-

-

[9]

2

2.92

C5H9NO4

146.0466

56.0548[M−H−CO2−CO−H2O], 84.0464[M−H−CO2−H2O]

4.60

Glutamic acid1

+

+

+

+

+

+

+

+

[10]

3

3.03

C6H12O7

195.0543

59.0177, 75.0119, 129.0218, 177.0436[M−H−H2O]

0.36

D-Mannonic acid

+

+

+

+

+

+

+

+

[11]

4

3.47

C5H10O5

149.0486

73.0329[M−H−H2O−CO2−CH2], 89.0273[M−H−H2O−C3H6], 105.0218[M−H−CO2]

4.36

2-Deoxypentonic acid

+

+

+

+

+

+

+

+

[12]

5

3.76

C4H6O5

133.0176

71.0165[M−H−H2O−CO2], 115.0060[M−H−H2O]

4.98

Malic acid

+

-

+

+

+

+

+

+

[13]

6

4.87

C30H26O14

609.1355

177.0235[M−H−C15H12O6−C6H8O3],  305.0714[M−H−C15H12O6], 423.0776[M−H−C8H10O5]

-1.62

Prodelphinidin B1

+

-

-

-

+

-

-

-

[12]

7

5.35

C7H6O5

169.0143

69.0390[M−H−CO2−2CO], 97.0322[M−H−CO2−CO],  107.0141[M−H−CO2−H2O], 125.0248[M−H−CO2]

3.25

Gallic acid1

+

+

+

+

+

+

+

+

[14]

8

5.74

C30H26O12

577.1444

  125.0277[1,4A], 179.0721[M−H−C15H12O6−C6H5O2], 245.0743[M−H−C15H12O6−CO2], 289.0749[M−H−C15H12O6], 407.0931[M−H−C8H8O3−H2O], 425.0829[M−H−C8H8O3], 451.0986[M−H−H2O−C6H5O2], 559.1335[M−H−H2O]

-1.65

Procyanidin B1

+

+

+

+

-

+

+

+

[15]

9

6.04

C9H11NO2

164.0723

103.0595[M−H−NH3], 147.0471[M−H−NH3−CO2]

1.71

Phenylalanine1

+

+

+

+

+

+

+

+

[16]

10

6.15

C7H6O4

153.0199

81.0361[M−H−CO−CO2], 91.0197[M−H−CO2−H2O],  109.0298[M−H−CO2], 125.0323[M−H−CO]

1.70

Protocatechuic acid1

+

+

+

+

+

+

+

+

[17]

11

6.66

C15H14O7

305.0661

125.0275[1,4A], 137.0266[1,3A], 167.0339[1,2A]

-0.95

Epigallocatechin

+

+

+

+

+

+

+

+

[18]

12

6.70

C16H18O9

353.0868

135.0478[M−H−C7H10O5−CO2], 179.0391[M−H−C7H10O5],  191.0596[M−H−caffeoyl]

-2.86

Neochlorogenic acid1

+

+

+

+

+

+

+

+

[19]

13

7.17

C7H6O5

169.0146

83.0170[M−H−CO2−2H2O], 125.0271[M−H−CO2],  151.0064[M−H−H2O]

2.07

2,3,4-Trihydroxybenzoic acid

+

+

+

+

+

+

+

+

[20]

14

7.24

C13H14O12

361.0108

125.0278, 151.0065, 169.0177

-1.11

2-O-Galloylgalactaric acid

+

+

+

+

+

+

+

+

[11]

15

8.22

C16H18O9

353.0872

85.0311, 161.0231[M−H−C7H10O5−H2O],  179.0391[M−H−C7H10O5], 191.0596[M−H−caffeoyl]

-1.73

Chlorogenic acid1

+

+

+

+

+

+

+

+

[21]

16

8.67

C27H32O15

595.1673

269.0876[M−H−RG−H2O], 287.0995[M−H−RG]

0.77

Eriocitrin

+

+

+

+

+

+

+

+

[22]

17

9.97

C21H22O11

449.1063

269.0498[M−H−Glc−H2O], 287.0599[M−H−Glc]

-4.16

Marein

+

+

-

+

+

+

+

+

[23]

18

10.29

C9H6O4

177.0221

121.0327[M−H−2CO], 149.0269[M−H−CO]

3.78

6,7-Dihydroxycoumarin

+

+

+

+

+

+

+

+

[12]

19

10.49

C9H8O4

179.0351

89.0413[M−H−CO2−CO−H2O], 109.0440[M−H−CO2−CO], 135.0449[M−H−CO2]

0.22

Caffeic acid1

+

+

+

+

+

+

+

+

[21]

20

11.37

C9H10O5

197.0460

123.0073[M−H−C3H6O2], 167.0022[M−H−CH2O]

2.28

Syringic acid1

+

+

+

+

+

+

+

+

[24]

21

11.60

C13H12O8

295.0464

115.0060, 133.0167

1.56

Caffeoylmalic acid

+

+

+

+

+

-

-

+

[11]

22

12.65

C15H10O5

269.0564

107.0192[1,3A−CO2], 117.0345[M−H−C7H4O4], 151.0072[1,3A], 225.0652[M−H−CO2]

4.70

Apigenin

+

+

+

-

-

+

+

-

[21]

23

13.52

C26H28O14

563.1391

383.0761[M−H−2C3H6O3], 443.0970[M−H−C4H8O4],  473.1079[M−H−C3H6O3]

-2.70

Schaftoside1

+

+

+

+

+

+

+

+

[25]

24

14.39

C33H40O20

755.2041

255.0334, 271.0286, 301.0402[M−H−RG−Rha]

0.11

Quercetin 3-O-beta-robinoside 7-O-alpha-L-rhamnopyranoside

+

+

+

+

+

+

+

+

[11]

25

15.05

C9H8O3

163.0405

93.0356[M−H−C3H2O2], 119.0508[M−H−CO2]

2.64

P-coumaric acid1

+

+

+

+

+

+

+

+

[26]

26

15.23

C21H20O13

479.0825

271.0294, 287.0249, 317.0366[M−H−Gal]

-1.27

Myricetin-3-Galactoside

+

+

+

+

+

+

+

+

[12]

27

15.59

C32H38O20

741.1854

179.0022[M−H−RG−Xyl−H2O], 301.0404[M−H−RG−Xyl]

-1.82

Quercetin 3-O-xylosyl-rutinoside

+

+

+

+

+

+

+

+

[11]

28

15.8

C10H10O4

193.0507

133.0319[M−H−C2H4O2]

0.71

Ferulic acid1

+

+

+

+

+

+

+

+

[27]

29

16.12

C21H20O10

431.0986

311.0594[M−H−C4H8O4], 341.1187[M−H−C3H6O3]

0.53

Vitexin1

+

+

-

+

+

+

+

+

[28]

30

16.18

C26H28O14

563.1393

383.0765[M−H−2C3H6O3], 443.0971[M−H−C4H8O4]

-3.60

Isoschaftoside1

+

+

+

-

+

+

+

+

[21]

31

16.76

C33H40O19

739.2051

227.0382, 255.0329, 285.0431[M−H−RG−Rha]

-3.66

Kaempferol 3-rutinosyl 7-O-alpha-L-rhamnoside

+

+

+

+

+

+

+

+

[12]

32

17.56

C21H20O10

431.0982

269.0495[M−H−Rha], 311.0590[M−H−C4H8O4],  341.0688[M−H−C3H6O3]

0.39

Isovitexin1

+

+

-

+

+

+

+

+

[25]

33

17.97

C27H30O15

593.1495

285.0393[M−H−RG]

-2.85

Kaempferol 3-O-rutinoside1

+

+

+

+

+

+

+

+

[29]

34

18.65

C21H20O12

463.0872

301.0408[M−H−Gal]

-2.16

Hyperoside1

+

+

+

+

+

+

+

+

[21]

35

18.82

C27H30O16

609.1461

151.0035[1,3A], 301.0353[M−H−RG]

-0.02

Rutin1

+

+

+

+

+

+

+

+

[21]

36

18.94

C21H20O12

463.0887

301.0498[M−H−Glc]

-2.26

Isoquercetin1

+

+

+

+

+

+

+

+

[21]

37

21.22

C21H20O10

431.0963

151.0073[1,3A], 269.0489[M−H−Rha], 413.2366[M−H−H2O]

-4.80

Afzelin

+

+

+

+

+

+

+

+

[30]

38

21.74

C21H20O11

447.0902

151.0070[1,3A], 285.0447[M−H−Gal]

-3.17

Kaempferol 3-O-galactoside1

+

+

+

+

+

+

+

+

[31]

39

21.84

C21H20O11

447.0901

301.0402[M−H−Rha]

-3.65

Quercitrin

+

+

+

+

+

+

+

+

[32]

40

22.76

C21H20O11

447.0894

151.0065[1,3A], 285.0450[M−H−Glc]

-3.80

Astragalin1

+

+

+

+

+

+

+

+

[33]

41

23.33

C10H18O4

201.1172

111.0852[M−H−CO2−H2O−CO], 139.1161[M−H−CO2−H2O], 183.1056[M−H−H2O]

4.17

3-Methylazelaic acid

+

+

+

+

+

+

+

+

[11]

42

26.35

C15H10O6

285.0404

227.0409[M−H−2CHO]

0.10

Luteolin

+

+

+

+

+

-

-

+

[28]

43

26.38

C16H12O6

299.0561

284.0307[M−H−CH3]

-0.21

Kaempferide

+

-

-

+

-

+

+

-

[21]

44

27.21

C15H10O7

301.0348

107.0172[0,4A], 121.0483[M−H−1,3A], 151.0034[1,3A],  179.0341[1,2A], 193.0287[M−H−B ring], 257.0401[M−H−CO2], 273.0393[M−H−CO]

-1.93

Quercetin1

+

+

+

+

+

+

+

+

[21]

45

28.34

C15H10O6

285.0401

133.0295[1,3A−H2O], 151.0044[1,3A], 162.8200[0,2A],

-1.22

Kaempferol1

+

+

+

+

+

+

+

+

[21]

46

28.43

C15H12O5

271.0610

119.0529[M−H−C7H4O4], 151.0073[1,3A]

0.61

Naringenin

+

+

-

-

+

-

+

-

[30]

Note: “1” comparison with reference standard.

Point 2 Line 151. What is the standard? Kaempferol?

Reply 2 Thanks for your kind advice and we are very sorry for our negligence. Compound 43 is identified as kaempferide compared with references, not standard. We have revised this sentence in Line no 147, Page number 9 of 21, and revised it as follows: “ Compared with references, it can be tentatively identified as kaempferide.”

Point 3 Line 269. After “the samples of LH from 8 habitats” show the legend of each habitat “S” (Bazhong, Guangyuan, Yibin, Zigong, Deyang, Suining, Leshan, and Nanchong), for example, S1, xx; S2, xx; S3, xxx and remove the habitat name in other text.

Reply 3 Thanks for your kind advice. The revised part was now presented in Line no 283, Page number 15 of 21, revised as follows: “ PCA was used to analyze the differences between the samples of LH from different habitats (S1, Bazhong; S2, Guangyuan; S3, Yibin; S4, Zigong; S5, Deyang; S6, Suining; S7, Leshan; S8, Nanchong) and the correlation between samples.”

Point 4 Figure 5. Please circle only the habitat clusters, removing the cycle in each substance. Centralize the figure.

The PCA and PLS models need to be improved. Please, try to use S-plot and VIP in the V graph to better explain the values.

Reply 4 Thanks for your professional commend. 

  • PCA has been modified to divide eight habitats into four sets according to the axis direction. The revised PCA diagram is shown below (Figure 9). And we have put this revised PCA diagram in Line no 291, Page number 15 of 21.

Figure 9. PCA scores plot of LH samples from different habitats.

  • Thanks for your professional commend.Due to the software limitation, the S-plot diagrams in SIMCA-P 13.0 software can only be displayed on the basis of the OPLS-DA. In view of your suggestion, we have applied PLS-DA and OPLS-DA respectively to screen for chemical components with VIP > 1. The results show that the difference is not significant. PLS-DA produced ideal results when we selected the analysis method in the beginning, so we finally chose PLS-DA to screen out 4 common differential chemical constituents from samples in Sichuan Basin. Therefore, we finally adopt the PLS-DA, and the results of LH samples from different habitats in Sichuan Basin are also shown in Line no 303, Page number 17 of 21 in the paper.

We have checked and corrected other errors in the full text.

Once again, thank you very much for your comments and suggestions.

Round 2

Reviewer 2 Report

The revised version is suitable for publication in Molecules